# Blood Count-Derived Inflammatory Markers Correlate with Lengthier Hospital Stay and Are Predictors of Pneumothorax Risk in Thoracic Trauma Patients

**DOI:** 10.3390/diagnostics13050954

**Published:** 2023-03-02

**Authors:** Vlad Vunvulea, Răzvan Marian Melinte, Klara Brinzaniuc, Bogdan Andrei Suciu, Adrian Dumitru Ivănescu, Ioana Hălmaciu, Zsuzsanna Incze-Bartha, Ylenia Pastorello, Cristian Trâmbițaș, Lucian Mărginean, Réka Kaller, Ahmad Kassas, Timur Hogea

**Affiliations:** 1Doctoral School of Medicine and Pharmacy, George Emil Palade University of Medicine, Pharmacy, Sciences and Technology of Targu Mures, 540142 Targu Mures, Romania; 2Department of Radiology, Mures County Emergency Hospital, 540136 Targu Mures, Romania; 3Department of Anatomy, George Emil Palade University of Medicine, Pharmacy, Science and Technology of Targu Mures, 540139 Targu Mures, Romania; 4Department of Orthopedics, Humanitas MedLife Hospital, 400664 Cluj Napoca, Romania; 5Clinic of Vascular Surgery, Mures County Emergency Hospital, 540136 Targu Mures, Romania; 6Faculty of Medicine in English, George Emil Palade University of Medicine, Pharmacy, Science and Technology of Targu Mures, 540139 Targu Mures, Romania

**Keywords:** thoracic trauma, chest injury, pneumothorax, inflammatory markers, neutrophil-to-lymphocyte ratio, monocyte-to-lymphocyte ratio, platelet-to-lymphocyte ratio, systemic inflammatory index, systemic inflammatory response index, aggregate inflammatory systemic index

## Abstract

(1) Background: Trauma is one of the leading causes of death worldwide, with the chest being the third most frequent body part injured after abdominal and head trauma. Identifying and predicting injuries related to the trauma mechanism is the initial step in managing significant thoracic trauma. The purpose of this study is to assess the predictive capabilities of blood count-derived inflammatory markers at admission. (2) Materials and Methods: The current study was designed as an observational, analytical, retrospective cohort study. It included all patients over the age of 18 diagnosed with thoracic trauma, confirmed with a CT scan, and admitted to the Clinical Emergency Hospital of Targu Mureş, Romania. (3) Results: The occurrence of posttraumatic pneumothorax is highly linked to age (*p* = 0.002), tobacco use (*p* = 0.01), and obesity (*p* = 0.01). Furthermore, high values of all hematological ratios, such as the NLR, MLR, PLR, SII, SIRI, and AISI, are directly associated with the occurrence of pneumothorax (*p* < 0.001). Furthermore, increased values of the NLR, SII, SIRI, and AISI at admission predict a lengthier hospitalization (*p* = 0.003). (4) Conclusions: Increased neutrophil-to-lymphocyte ratio (NLR), monocyte-to-lymphocyte ratio (MLR), platelet-to-lymphocyte ratio (PLR), systemic inflammatory index (SII), aggregate inflammatory systemic index (AISI), and systemic inflammatory response index (SIRI) levels at admission highly predict the occurrence of pneumothorax, according to our data.

## 1. Introduction

Trauma is the world’s top cause of disability and death in the first four decades of life. In this age group, the number of young adults who die from trauma exceeds all deaths from cancer combined [1]. Thoracic injuries are highly significant in patients with severe trauma, occurring in up to 50% of patients with polytrauma [2]. According to the current literature, the mortality rate following thoracic trauma varies between 25 and 50%, depending on the associated injuries [3,4]. 

The assessment of thoracic trauma severity determines the choice of first therapy and the subsequent clinical course when treating patients with polytrauma. Although thoracic trauma specifically has received little attention in the literature, there is a wealth of information on the mortality-associated risk factors following trauma in general [5,6]. 

Pathological inflammatory and anti-inflammatory responses that occur in the first hours following extensive trauma are one of the major contributing factors to mortality in post-traumatic patients and remain challenging to control and distinguish from a physiological immune reaction [7]. The balance between these two antagonistic inflammatory responses, as predictors of outcomes in trauma patients, has received a lot of attention recently. In response to severe injury, patients frequently experience a variety of anomalies in their host defense mechanisms [8]. Systemic inflammatory response syndrome (SIRS) is the result of an unbalanced inflammatory response that escalates and releases an excessive amount of inflammatory mediators, such as IL-1, IL-6, IL-8, and TNF [9]. The injury burden is increased by the progression of such an uncontrolled cytokine cascade and hyperinflammation. This can lead to detrimental and frequently fatal events such as SIRS and multiple organ dysfunction syndrome (MODS) [10]. Recently, there has been a growing interest in developing a trustworthy biomarker that can assess the prognosis of patients with thoracic trauma [11].

The neutrophil-to-lymphocyte ratio (NLR) is one of the most accessible markers. This ratio has been proven to significantly predict the outcomes of patients with COVID-19 infection [12,13,14,15], cardiovascular diseases [16,17,18,19,20], and kidney disease [12,21] and oncology [22,23,24]. Another well-studied biomarker is the platelet-to-lymphocyte ratio (PLR), which has been shown to have excellent predictive value for the prognosis of patients in the fields of orthopedics [19,25,26] and trauma care [27,28,29,30]. Based on routine blood tests at admission, several other ratios can be calculated, such as the monocyte-to-lymphocyte ratio (MLR), aggregate inflammatory systemic index (AISI), systemic inflammatory response index (SIRI), and systemic inflammatory index (SII). 

The monocyte-to-lymphocyte ratio (MLR) has been proven to be a valid predictor of the occurrence of complication in strokes [31], and the outcomes and severity of hematological disorders [32] and oncological patients [33]. 

The NLR, PLR, and MLR, for example, have been the subject of a growing number of studies in recent years. However, their findings suggest that a combination of these ratios would increase their predictive value [34,35,36]. Thus, the aggregate inflammatory systemic index (AISI), systemic inflammatory response index (SIRI), and systemic inflammatory index (SII) were discovered and proven useful when evaluating the severity and prognosis of patients with various chronic and acute pathologies [37,38,39]. 

The prognosis ratios calculated from routine blood tests appear to be a helpful and cost-effective resource in trauma management. Although there are mentions in the literature of the correlation between the NLR and the outcomes of thoracic trauma patients [11], there are few to no papers published regarding the use of the PLR, MLR, SII, AISI, and SIRI as prognostic factors for the outcomes of patients with thoracic trauma.

The purpose of this study is to establish the prognostic value of inflammatory biomarkers and the underlying risk factors in patients with thoracic trauma.

## 2. Materials and Methods

### 2.1. Study Design

The present study was designed to be an observational, retrospective, analytical cohort study where we included all patients over the age of 18 who presented, were diagnosed with thoracic trauma, and admitted to the County Emergency Clinical Hospital of Targu Mureş, Romania, between January 2015 and December 2022. All patients included in our study underwent a radiological examination of either a conventional X-ray or a CT scan, and all were diagnosed with thoracic trauma as the main diagnosis. We excluded patients who passed away within the first 24 h, suffered severe bone fractures with need for specialized orthopaedical care, had a history of hematological or oncological disorders, presented thromboembolic events in the last two months, and patients with pneumonia. We also excluded patients suffering from mediastinal hematoma and aortic dissection as such patients are referred to the cardiovascular surgery department, not the thoracic surgery department. All patients included in our study suffered from peacetime injuries.

We initially split the patients in two categories: “Pneumothorax” and “No Pneumothorax” based on the findings at admission. 

### 2.2. Data Collection

We collected the following data from our patients: age, sex, medical history (of diabetes mellitus—DM, arterial hypertension—AH, atrial fibrillation—AF, ischemic heart disease—IHD, myocardial infarction—MI, chronic obstructive pulmonary disease—COPD, peripheral arterial disease—PAD, chronic kidney disease—CKD, tobacco use, and obesity (BMI > 30)) and length of hospital stay (LOS). Moreover, we were interested in the routine blood tests at admittance. From these results, we extracted the following data: hemoglobin levels, hematocrit, neutrophil count, monocyte count, lymphocyte count, platelet count, sodium, and potassium. We were also interested in the number and location of rib fractures. All data were collected from the hospital’s integrated electronic database. 

### 2.3. Inflammatory Biomarkers 

From the results of the initial blood test at admittance, we managed to calculate the following ratios: MLR = monocytes/lymphocytesNLR = neutrophils/lymphocytesPLR = platelets/lymphocytesSII = (neutrophils × platelets)/lymphocytesSIRI = (monocytes × platelets)/lymphocytesAISI = (neutrophils × monocytes × platelets)/lymphocytes

### 2.4. Study Outcomes

The primary endpoint for our study was the risk of pneumothorax development. We also recorded the length of hospital stay as an outcome, making it our secondary endpoint. 

### 2.5. Statistical Analysis

Software-wise, we used SPSS for Mac OS (28.0.1.0) (SPSS, Inc., Chicago, IL, USA). All systemic inflammatory marker associations with category factors were evaluated using chi-square tests, whilst differences in continuous variables were evaluated using Student *t*-tests or Mann–Whitney tests. The receiver operating characteristic (ROC) curve analysis was used to determine the cut-off values for inflammatory markers and evaluate their predictive potential. Based on the Youden index (Youden index = sensitivity + specificity 1, ranging from 0 to 1), the suitable NLR, MLR, PLR, SII, SIRI, and AISI cut-off values were determined using the ROC curve analysis. 

## 3. Results

During our study period, we identified 611 patients suffering from thoracic trauma that met the inclusion criteria for our study. The mean age was 47.48 ± 18.66 (18–98) (Table 1). The majority of patients included were males (448, 73.32%), with 114 (25.44%) of them suffering from pneumothorax at admission. At admission, 155 patients (25.37%) presented with pneumothorax. The mean length of hospital stay was 6.73 ± 4.14 days.

After splitting the patients into two lots depending on the occurrence of pneumothorax, we noticed an increase in the mean age for the “Pneumothorax” group to 51.68 ± 19.39 (*p* = 0.002), as well as a higher incidence of tobacco use (*p* = 0.019) and obesity (*p* = 0.038). As for the etiology of trauma, we found the majority of patients suffered from blunt trauma (539/611 patients, 88.22%). In this category, we considered all patients who suffered from motor vehicle accidents, workplace accidents, accidental falls, sport-related injuries, and suicide attempts. In terms of patients who experienced penetrating trauma, we included all patients who experienced hetero-aggression and stabbings. They accounted for 11.78% of all patients and 41.93% of pneumothorax patients. Moreover, patients who suffered from posttraumatic pneumothorax showed higher sodium levels (*p* = 0.024), higher neutrophil (*p* < 0.0001), monocyte (*p* < 0.0001), and platelet (*p* = 0.009) counts, and lower lymphocyte (*p* < 0.0001) counts. All hematological ratios were higher in the “Pneumothorax” group (*p* < 0.0001). The length of hospital stay was also longer in the “Pneumothorax” group (*p* = 0.003). 

The receiver operating characteristic curves of all hematological ratios were computed in order to assess if the initial values of these indicators were predictive for the occurrence of pneumothorax in patients with thoracic injuries (Figure 1). Table 2 displays the optimal cut-off value calculated using Youden’s index, the areas under the curve (AUC), and the prediction accuracy of the markers.

In terms of systemic inflammatory makers and the length of hospital stay, we computed the Spearman correlation, and we identified a positive correlation between the NLR, SII, SIRI, and AISI and length of hospital stay (all *p* < 0.05), as highlighted in Figure 2.

We proceeded with the multivariate analysis of age, risk factors, all inflammatory ratios, and the occurrence of pneumothorax within the patients in the second group, as shown in Table 3. Furthermore, older patients (OR:1.01, *p* = 0.02), the presence of COPD (OR:2.93, *p* = 0.02), as well as tobacco (OR:2.20, *p* = 0.01), act as predictive factors for pneumothorax risk. In contrast, obesity acts as protective factor against pneumothorax (OR:0.65, *p* = 0.03). We considered an increased value of the NLR as being a value higher than the identified cut-off (NLR > 6, *p* < 0.001). This is similar for a high MLR (MLR > 0.62, *p* < 0.001), PLR (PLR > 165.71, *p* < 0.001), SII (SII > 1632.86, *p* < 0.001), SIRI (SIRI > 6.17, *p* < 0.001), and AISI (AISI > 1479, *p* < 0.001). 

## 4. Discussion

According to the recent literature, thoracic trauma is a frequently occurring presentation in injured patients [40]. Post-traumatic pneumothorax is a common complication of chest injuries, occurring in between 20 and 55% of patients, associated with relatively high morbidity and mortality. The mean age reported in the literature varies between 39 and 61 years old [41,42,43,44,45].

However, it is a preventable cause of death. Early diagnosis of pneumothorax can aid in the management of such patients, prevent hemodynamic deterioration, or occurrence of other complications. 

In the present study, the incidence of pneumothorax was 25.37% (*n* = 155/611), with a mean age of 47.48 ± 18.66, are similar findings to those found in the literature. 

Most studies found in the recent literature report a negative impact on the outcomes of trauma patients among smokers [46,47,48]. In spite of all these findings, a recent paper published by Grigorian et al., which included 282,986 patients with chest injuries, reports a significantly better outcome in smokers, with a lower number of ventilator days (*p* = 0.009) and a lower rate of in-hospital mortality (*p* < 0.001). However, smokers appear to develop a higher rate of pneumonia (*p* < 0.001) [49]. In our study, we identified a total of 34 chronic tobacco users (5.56%) and identified smoking as a negative predictor of outcomes, with a higher incidence of pneumothorax occurrence (OR = 2.29, *p* = 0.01). A plausible reason for this discrepancy can be attributed to the high proportion of smokers included in the study of Grigorian et al. totaling 57,619 patients (20.4%). 

The role of obesity as a risk factor for the outcomes of trauma patients is a topic of debate in the current literature. There are plenty of papers, including complex meta-analyses, that advocate for poorer outcomes of obese patients following major trauma [50,51,52,53]. Some papers, however, found that obese patients suffering from trauma have a more favorable outcome with a faster recovery [54,55]. According to our findings, obesity is a protective factor for the development of pneumothorax in patients suffering from chest injuries (OR = 0.65, *p* = 0.003). One of the reasons for such paradoxical findings can be attributed to the protective role of the adipose tissue upon blunt chest injuries. 

The type of trauma appears to also play an important role in the development of pneumothorax. We notice that the majority of patients included in our study suffered from blunt chest injuries, which is to be expected as we did not have any wartime injuries reported in the past few years. We also notice that the majority of patients with penetrating trauma develop pneumothorax (65/72), but as the number of patients suffering from penetrating trauma is low, we can consider these data as purely observational.

The predictive values of hematological ratios in trauma patients have reportedly been researched more and more, although with conflicting results. Additionally, there has been a significant rise in the need for prognostic tools in trauma patients with unfavorable evolution and decompensation.

Our study included 611 patients diagnosed with thoracic trauma. We identified the inflammatory biomarkers in patient blood samples at admission and determined the presence of pneumothorax using CT scans at admission. Our study’s most important outcome is that the high baseline values for the NLR, MLR, PLR, AISI, SII, and SIRI are strong predictors for the development of post-traumatic pneumothorax. To the best of our knowledge, this is the first study to demonstrate that patients with high hematological ratios were more likely to develop pneumothorax and that the ratios predict a longer hospital stay. 

According to Soulaiman et al., there is a statistically proven association between the NLR at admission and the outcomes of trauma patients, where a higher NLR predicts an unfavorable outcome [8]. According to this study, the optimal cut-off value for the NLR at admission was 4, which is a close value to our findings, with an AUC = 0.63 (70.3% sensitivity and 56.4% specificity), highlighting a satisfactory test quality. In comparison, we computed a cut-off value for the NLR of 6, with an AUC = 0.79, highlighting an increased test quality. 

In contrast, other studies, such as the one conducted by Dilektasli et al., revealed no statistically significant association between the NLR calculated from the blood samples at admission and the outcomes of trauma patients [56]. 

These controverted findings inspired another study, conducted by Younan et al. [57], to investigate the association between the NLR and the outcomes of trauma patients. According to the aforementioned, an increasing trajectory of the NLR (calculated at admission, and 24 and 48 h later) is strongly associated with the outcomes of the patients (*p* = 0.002) and length of hospital stay (*p* < 0.001). The total number of patients included in their study appears to be more modest (207 patients); patients with all types of trauma were included, not just chest injuries. Despite all these limitations, the findings of their study appear to support ours. 

According to Jo et al., the PLR has significant prediction power for the outcomes of trauma patients (*p* < 0.0001) [27]; however, they found a higher lymphocyte count in the non-survival group compared to the survival group (183.0 [141.0;230.0] vs. 227.0 [188.0;265.0]). The PLR was also lower in the non-survival group compared to the survival group (51.3 [32.3;77.9] vs. 124.2 [79.5;187.2]). These findings are contrary to ours, where the lower the lymphocyte count and the higher the PLR, the worse the outcome. 

A recent study by Rau et al. [58], including 479 trauma patients, found that comorbidities and hematological ratios (NLRs, MLRs, and PLRs) do not possess any predicting capabilities in the outcomes of such patients. Although some of their findings appear to contradict ours, we must remember that their study included all types of trauma and survival was considered as the final outcome of patients. The fact that a majority of the patients included in their study had suffered from a head or neck injury can be an explanation for their findings. Another reason for the lack of association between the hematological ratios and the outcomes of trauma patients can be attributed to the selection criteria. Their study also included patients who underwent invasive procedures, such as surgery, or patients who required resuscitation or blood transfusion, which are factors that can alter hematological ratios. We have taken into account these possible limitations of such reputable studies; this is the reason why our study’s main focus was thoracic trauma, with specific exclusion criteria. 

In the current study, according to the multivariate analysis, all the hematological ratios were able to predict the occurrence of pneumothorax (*p* < 0.0001 in all cases). Moreover, we proved that some increased hematological ratios can indirectly predict the occurrence of complications through an increased length of hospital stay (SII *p* = 0.022, r = 0.093; SIRI *p* = 0.008, r = 0.108; and AISI *p* = 0.009, r = 0.106). Lastly, the present paper also revealed a major risk factor for traumatic pneumothorax development in tobacco use (OR = 2.29, *p* = 0.019), whilst obesity is a protective factor (OR = 0.65, *p* = 0.038). 

The findings of our previous studies on the role of hematological biomarkers as predictive factors in the outcomes of both specific splenic trauma [29] and abdominal trauma [30] support the findings of the current paper. In the first paper, we found a significant association between the NLR and the severity of splenic injury (*p* = 0.02). The findings of the second paper revealed that the NLR, PLR, MLR, AISI, SII, and SIRI are powerful predictors of the development of acute kidney injury, mortality, and a composite endpoint of these two outcomes in abdominally injured patients (*p* < 0.001 in all cases).

Nevertheless, the present study has a set of limitations. The first limitation relies on the design of the study as a retrospective monocentric study. Further improvement could be brought by extending the research to a multicentric prospective study. Secondly, due to the retrospective nature of our study, we were unable to gather enough data on chronic medications administered before admission (corticosteroids or anti-inflammatory drugs), which prevented us from assessing how various medications affect inflammatory biomarkers. Lastly, the study only analyzed the inflammatory biomarkers at admission. Repeated determination throughout the hospitalization period may better reflect the dynamics of the inflammatory process and may improve the quality of our findings. 

In spite of all these limitations, we consider our findings to be a stepping stone toward the development of new risk scoring systems for the improvement of the overall management of thoracic trauma patients and the early identification of patients at risk. We consider these hematological ratios to be especially important, taking into account their ease of determination and the low cost of assessment.

## 5. Conclusions

Our data show that patients with thoracic injuries, who have elevated NLRs, PLRs, MLRs, SIIs, SIRIs, and AISIs at admission at values that are above our calculated cutoff, are likely to have sustained severe thoracic trauma, are likely to have developed pneumothorax, and will likely follow a long evolution with a long duration of hospitalization. Additionally, we proved that tobacco use is a strong predictor of the development of post-traumatic pneumothorax in such patients, whilst obesity is a protective factor. 

Given the ease of use of such ratios and the low cost of these metrics, they can be used in clinical practice to categorize patient treatment groups, develop predictive patterns, and classify risk groups for admission.

## Figures and Tables

**Figure 1 diagnostics-13-00954-f001:**
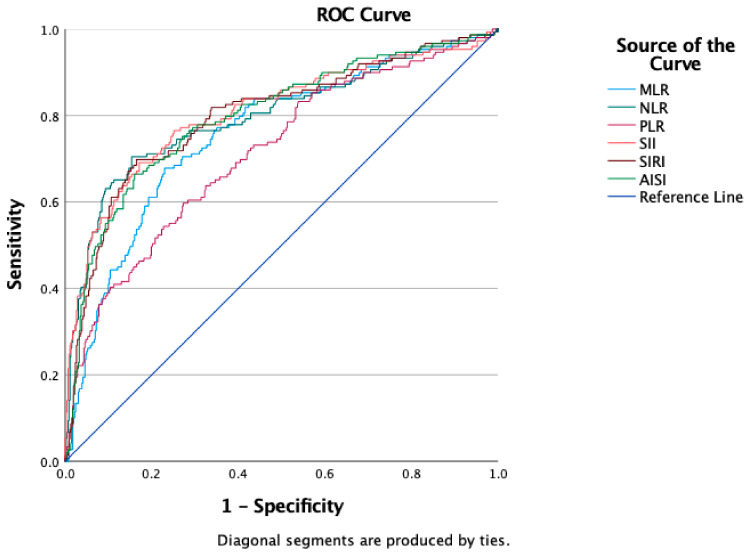
ROC curve analysis concerning the pneumothorax risk: NLR (AUC: 0.798; *p* < 0.0001), MLR (AUC: 0.758; *p* < 0.0001), PLR (AUC: 0.714; *p* < 0.0001), SII (AUC: 0.807; *p* < 0.0001), SIRI (AUC: 0.799; *p* < 0.0001), and AISI (AUC: 0.799; *p* < 0.0001).

**Figure 2 diagnostics-13-00954-f002:**
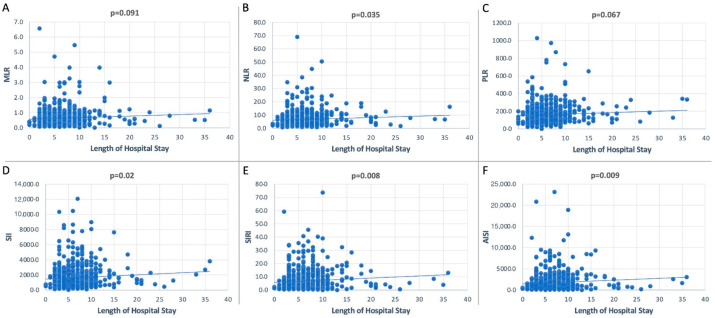
Plot representation of the dispersion of data of the association between length of hospital stay and the inflammatory biomarkers.

**Table 1 diagnostics-13-00954-t001:** Demographic information, risk factors, comorbidities, laboratory data, and outcomes were compiled. The patients were divided into two lots based on the presence of pneumothorax at admission.

Variables	All Patients *n* = 611	Pneumothorax *n* = 155 (25.37%)	No Pneumothorax *n* = 456 (74.63%)	*p* Value
Age mean ± SD (min–max)	47.48 ± 18.66 (18–98)	51.68 ± 19.39 (19–93)	46.12 ± 18.23 (18–98)	0.002
Male	448 (73.32%)	114 (73.54%)	334 (73.24%)	0.94
Female	163 (26.67%)	41 (25.15%)	122 (26.75%)	
Comorbidities and risk factors				
Arterial hypertension, No. (%)	160 (10.8%)	40 (25.81%)	120 (26.32%)	0.828
Ischemic heart disease, No. (%)	82 (13.42%)	24 (15.48%)	58 (12.72%)	0.269
Atrial fibrillation, No. (%)	33 (5.4%)	8 (5.16%)	25 (5.48%)	0.984
Myocardial infarction, No. (%)	20 (3.27%)	6 (3.87%)	14 (3.07%)	0.262
Diabetes mellitus, No. (%)	66 (10.8%)	12 (7,74%)	54 (11.84%)	0.122
Chronic obstructive pulmonary disease, No. (%)	55 (9%)	23 (14.84%)	32 (7.02%)	0.066
Peripheral arterial disease, No. (%)	45 (7.36%)	12 (7.74%)	33 (7.24%)	0.712
Chronic kidney disease, No. (%)	27 (4.42%)	9 (5.81%)	18 (3.95%)	0.271
Tobacco, No. (%)	34 (5.56%)	15 (9.68%)	19 (4.17%)	0.019
Obesity, No. (%)	224 (36.66%)	44 (28.39%)	180 (39.47%)	0.038
Type of trauma				
Blunt	539 (88.22%)	90 (58.06%)	449 (98.4%)	<0.001
Penetrating	72 (11.78)	65 (41.93%)	7 (1.6%)	
Laboratory data				
Hemoglobin g/dL mean ± SD	12.54 ± 2.32	12.82 ± 2.01	12.45 ± 2.41	0.91
Hematocrit % mean ± SD	37.5 ± 7.15	38.29 ± 5.61	37.24 ± 7.56	0.12
Glucose mg/dL mean ± SD	132.46 ± 54.17	136.51 ± 47.49	131.15 ± 56.14	0.29
Sodium mean ± SD	137.31 ± 16.12	139.91 ± 12.71	136.48 ± 17	0.024
Potassium mean ± SD	4.38 ± 1.116	4.3 ± 0.88	4.4 ± 1.24	0.34
Neutrophils ×10^3^/μL mean ± SD	9.75 ± 5.07	13.13 ± 5.64	8.65 ± 4.35	<0.0001
Lymphocytes ×10^3^/μL mean ± SD	1.94 ± 1.02	1.52 ± 0.93	2.08 ± 1.01	<0.0001
Monocyte ×10^3^/μL mean ± SD	0.96 ± 0.86	1.18 ± 0.86	0.89 ± 0.84	<0.0001
Plt ×10^3^/μL mean ± SD	248.53 ± 90.46	265.39 ± 110.07	243.10 ± 82.57	0.009
MLR, mean ± SD	0.64 ± 0.63	0.96 ± 0.66	0.53 ± 0.59	<0.0001
NLR, mean ± SD	6.76 ± 6.4	12.12 ± 9.23	5.04 ± 3.83	<0.0001
PLR, mean ± SD	163.45 ± 113.55	229.41 ± 155.68	142.18 ± 86.14	<0.0001
SII, mean ± SD	1636.60 ± 1554.68	3040.74 ± 2251.73	1183.75 ± 853.06	<0.0001
SIRI, mean ± SD	6.89 ± 8.14	13.25 ± 10.4	4.83 ± 5.97	<0.0001
AISI, mean ± SD	1739.16 ± 2372.9	3421.47 ± 3097.28	1196.59 ± 1777.91	<0.0001
Outcomes				
Length of hospital stay, mean ± SD	6.73 ± 4.14	7.62 ± 4.26	6.45 ± 4.06	0.003

**Table 2 diagnostics-13-00954-t002:** ROC curves, ideal cut-off values, AUC, and prediction accuracy of inflammatory indicators in terms of outcomes.

Variables	Cut-Off	AUC	Std. Error	95% CI	Sensitivity	Specificity	*p* Value
Pneumothorax							
NLR	6	0.798	0.024	0.751–0.845	76.5%	71%	<0.0001
MLR	0.62	0.758	0.023	0.712–0.804	70.5%	72.9%	<0.0001
PLR	165.71	0.714	0.025	0.665–0.763	60.4%	71.6%	<0.0001
SII	1632.86	0.807	0.023	0.761–0.852	77.2%	73.4%	<0.0001
SIRI	6.17	0.799	0.023	0.754–0.843	71.8%	76.2%	<0.0001
AISI	1479.7	0.799	0.023	0.754–0.843	71.1%	75.8%	<0.0001

**Table 3 diagnostics-13-00954-t003:** Multivariate analyses of the age, risk factors, inflammatory ratios, and the occurrence of pneumothorax.

	Pneumothorax		
	OR	95% CI	*p* Value
Age	1.01	1.006–1.02	0.002
COPD	2.93	1.15–7.50	0.02
Obesity	0.65	0.44–0.97	0.03
Tobacco	2.29	1.12–4.66	0.01
High NLR	6.03	3.94–9.21	<0.001
High MLR	5.97	3.97–8.97	<0.001
High PLR	3.51	2.39–5.17	<0.001
High SII	8.78	5.68–13.59	<0.001
High SIRI	8.55	5.43–13.47	<0.001
High AISI	7.73	5.06–11.85	<0.001

## Data Availability

Not applicable.

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
