# Peer review of "Blood Count-Derived Inflammatory Markers Correlate with Lengthier Hospital Stay and Are Predictors of Pneumothorax Risk in Thoracic Trauma Patients"

_diagnostics, 2023, doi:10.3390/diagnostics13050954_

Round 1

Reviewer 1 Report

Thanks authors for the submission - minor amendments are needed:

You report on the NLR and PLR and associate this with prediction of pneumothorax risk. You include over 600 patients and the methods, limitations and discussion seem appropriate, however there are areas for improvement.

The changes required are mainly grammar and phrasing: A good review by an English first-language science writer is advised

1) Orthopedy should be "orthopedics" for example.

2) As per the SAGER guideline "gender" should change to "sex" as you presumably include male/female only (these numbers are not reported at all 

3) I think you can safely state that the clinical predictors for pneumothorax are true (age, obesity, tobacco, COPD), but a better way of stating the relevance of the biochemical findings is that the inflammatory response in patients with pneumothorax is more aggressive or exaggerated and this leads to a significant difference in the ratios under discussion. It is consequence not prediction. The reason I say this is that the injury induces the inflammation - it is not there prior to the injury.

4) You also did not present the general demographics of the patients in the tables - age yes, but not sex, nor shocked versus non-shocked on arrival (known to influence the inflammatory response) and the breakdown of blunt versus penetrating (there is work showing the inflammatory response in penetrating chest trauma is far less than with blunt) - see:

Groeneveld KM, Hietbrink F, Hardcastle TC, Warren BL, Koenderman L, Leenen LP. Penetrating thorax injury leads to mild systemic activation of neutrophils without inflammatory complications. Injury, 2014, 45(3): 522-527. DOI 10.1016/j.injury.2013.09.030

5) You should adjust the conclusion paragraph accordingly

Author Response

Dear reviewer,

We would like to thank you and the reviewers for your insightful comments, which have greatly helped us to improve the quality of our manuscript.

We would like to mention that the lines written in the response to the comments are from the tracked changes version of the reviewed manuscript. We have rewritten some paragraphs according to the all-reviewers comments.

Point 1&2. 
We had a native-language co-author review and correct the entire manuscript. We hope that it meets MDPI standards.

Point 3. 
Thank you very much for your comments. Some of the expressions were indeed misleading and changed the content of the paper ever so slightly. We agree that inflammation is induced by trauma. We just wanted to point out that, with both pneumothorax and inflammation being induced by injury, there might be a linear relationship between the two, where both outcomes are co-dependent. We addressed this issue in the revised version of the manuscript. 

Point 4. 
We are grateful for your recommendation. According to it, we have added the sex distribution and the cause of trauma. We did not include any shocked patients in our study since most of them receive blood transfusions and undergo surgery right away, factors that may have altered our data.

Point 5. 
According to your kind recommendations, we hope that the changes made to the "Conclusions" paragraph are appropriate.

Thank you very much for the time offered to take our paper into consideration.

We hope that these revisions are sufficient to make our manuscript suitable for publication in this special issue of MDPI Diagnostics and look forward to hearing from you at your earliest convenience.

Reviewer 2 Report

My comments regarding the manuscript are listed below.

1. Authors listed exclusion criteria as follows: delay of hospitalization more than first 24 hours, severe bone fractures with need for specialized orthopaedical care, history of hematological or oncological disorders, thromboembolic events in the last two months. What about mediastinal haematoma, aortic dissection, underlying pneumonia?

2. Please indicate in the text the kind of trauma in included patients. Was it fall from height, car accident (blunt trauma) or acute injury (stab wound, gun injury)? Is it about war or peacetime injury?

3. It is better to decipher the abbreviation right in the table, not below the table. Because it is hard to read the text in that way.

4. Table 6 has no OR and CI 95% in the last column as stated

5. In the discussion section authors stated that obesity is a risk factor for pneumothorax. However, according to statistical analysis obesity has a protective role (OR 0.65). Subsequently, there is a mismatch between conclusions and data obtained. Please, explain or correct

6. Please, specify in conclusions what kind of biomarkers are the risk factors for the pneumothorax

7. What is the value of high-NLR, high-MLR and so on? Please, indicate it

Author Response

Dear reviewer,

We would like to thank you and the reviewers for your insightful comments, which have greatly helped us to improve the quality of our manuscript.

We would like to mention that the lines written in the response to the comments are from the tracked changes version of the reviewed manuscript. We have rewritten some paragraphs according to the all-reviewers comments.

Point 1. 
We are grateful for your comment. We considered underlying pneumonia as an exclusion criteria but forgot to mention it.
However, due to the Romanian medical system, patients with aortic dissection and mediastinal hematoma are referred to the cardiovascular surgery department, not the thoracic surgery department. This is the main reason why we did not mention these exclusion criteria. 

Point 2. 
Thank you very much for the recommendation. We have included these details in the results section of Table 1 and on lines 191–197.

Point 3. 
Thank you very much for your comment. According to your recommendation, we have removed the abbreviations and opted for the full name of all comorbidities.

Point 4. 
We are unsure of your recommendation. We do not have a table 6, and we ensured that all statistical analyses included an OR and a 95% CI.

Point 5. 
Thank you very much for the comment. It was our mistake, and we have corrected it accordingly. 

Point 6. 
Thank you for your recommendation. We have corrected the conclusion accordingly.

Point 7. 
We refer to high-NLR as being a value above the optimal cutoff computed for our study group. We are grateful for your comment and have added the information under lines 226-229.

Thank you very much for the time offered to take our paper into consideration.

We hope that these revisions are sufficient to make our manuscript suitable for publication in this special issue of MDPI Diagnostics and look forward to hearing from you at your earliest convenience.

Round 2

Reviewer 2 Report

I would like to thank authors for accepting previous suggestions.

However, there is still a room for the manuscript improvement.

Authors explained that ‘due to the Romanian medical system, patients with aortic dissection and mediastinal hematoma are referred to the cardiovascular surgery department, not the thoracic surgery department’. I think that these pathologies need to be included in the article as exclusion criteria.

Authors made corrections in the first collumn of the Table 1. Now they should to delete explanation of the abbreviations below the table.

Author Response

Dear reviewer, 
Thank you very much for the comments.

Point 1:Authors explained that ‘due to the Romanian medical system, patients with aortic dissection and mediastinal hematoma are referred to the cardiovascular surgery department, not the thoracic surgery department’. I think that these pathologies need to be included in the article as exclusion criteria.
Response: Thank you very much for your remark. We have added these exclusion criteria in line 125-127. 

Point 2: Authors made corrections in the first collumn of the Table 1. Now they should to delete explanation of the abbreviations below the table.
Response: Thank you very much for the suggestion. We have removed the abbreviations accordingly. 

Thank you so much for taking into consideration our manuscript, for your valuable time and for your interest in reviewing our manuscript. Your comments had a great contribution in improving our manuscript.

Best regards,

Suciu Bogdan Andrei